# Monitoring Carbon Stock and Land-Use Change in 5000-Year-Old Juniper Forest Stand of Ziarat, Balochistan, through a Synergistic Approach

**Hamayoon Jallat [1], Muhammad Fahim Khokhar [1,\*], Kamziah Abdul Kudus [2], Mohd Nazre [2], Najam u Saqib [1], Usman Tahir [3] and Waseem Razzaq Khan [2,4,\*]**

[1] Institute of Environmental Science and Engineering, National University of Science and Technology, Islamabad 44000, Pakistan; hjallat.mses2017iese@student.nust.edu.pk (H.J.); nsaqib.mses2017iese@student.nust.edu.pk (N.u.S.)

[2] Department of Forest Science & Biodiversity, Faculty of Forestry and Environment, Universiti Putra Malaysia, UPM Serdang, Selangor 43400, Malaysia; kamziah@upm.edu.my (K.A.K.); nazre@upm.edu.my (M.N.)

[3] Department of Forest Sciences, Chair of Tropical and International Forestry, Faculty of Environmental Science, Technische Universität, 01069 Dresden, Germany; usman.tahir@mailbox.tu-dresden.de

[4] Institut Ecosains Borneo, Universiti Putra Malaysia, Bintulu Kampus, Sarawak 97008, Malaysia

\* Correspondence: fahim.khokhar@iese.nust.edu.pk (M.F.K.); khanwaseem@upm.edu.my (W.R.K.); Tel.: +92-51-9085-4308 (M.F.K.); +60-19-3557-839 (W.R.K.)

**Abstract:** The Juniper forest reserve of Ziarat is one of the biggest *Juniperus* forests in the world. This study assessed the land-use changes and carbon stock of Ziarat. Different types of carbon pools were quantified in terms of storage in the study area in tons/ha i.e., above ground, soil, shrubs and litter. The Juniper species of this forest is putatively called *Juniperus excelsa* Beiberstein. To estimate above-ground biomass, different allometric equations were applied. Average above ground carbon stock of the forest was estimated as 8.34 ton/ha, 7.79 ton/ha and 8.4 ton/ha using each equation. Average carbon stock in soil, shrubs and litter was calculated as 24.35 ton/ha, 0.05 ton/ha and 1.52 ton/ha, respectively. Based on our results, soil carbon stock in the Juniper forest of Ziarat came out to be higher than the living biomass. Furthermore, the spatio-temporal classified maps for Ziarat showed that forest area has significantly decreased, while agricultural and barren lands increased from 1988 to 2018. This was supported by the fact that estimated carbon stock also showed a decreasing pattern between the evaluation periods of 1988 to 2018. Furthermore, the trend for land use and carbon stock was estimated post 2018 using a linear prediction model. The results corroborate the assumption that under a business as usual scenario, it is highly likely that the *Juniperus* forest will severely decline.

**Keywords:** Juniperus; above-ground biomass; land-use; allometric equation; satellite remote sensing; land cover classification

## 1. Introduction

The present forest area of the world is 4.06 billion ha and it has lost 178 million ha of forest since 1990, presenting a decrease of 4.2% [1]. It is estimated that the world's forest area has decreased from 31.6% to 31% of the total Earth's land area. With approximately 5% of the land area under forest, Pakistan finds its place among the low-forested countries in the world [2–4]. Major forest types in Pakistan include coastal mangroves, riverine forests, sub-tropical scrub forests, moist temperate conifer forests, dry temperate conifer forests, and irrigated plantations including linear plantations [5]. Based on the 'Forestry Sector Master Plan of 1992', 4,200,000 ha (4.8%) of Pakistan has natural forest [6]. Compared to that, in 2015, FAO (Food and Agriculture Organization) reported 1.9% or about 1,472,000 ha of forest area in Pakistan. This translates to a loss 2,728,000 ha of the forest cover, which has also been documented by the Pakistan Forestry Outlook study in 2009 and the Forest Resource Assessment in 2015. Evidently, there is a declining trend in the forest cover while agricultural and urban areas are expanding [7].

The genus *Juniperus* is one of the biggest conifer genera with 52 species worldwide and is extensively spread over the northern temperate region [8]. Juniper forest of Ziarat, also known as the second-largest forest of juniper in the world, that covers an area of about 110,000 hectares and was declared a Biosphere Reserve in 2013 [9]. The area of this Juniper forest as per the working plan 1960 is 247,166 acres (100,025 ha) [10]. However, Akram et al. [11] calculation based on object-based image analysis, found that the area of the juniper forest of Ziarat is 53,092 ha in 2010. This discrepancy clearly shows that the juniper forest of Ziarat faced the threat of both deforestation and forest degradation [12]. Not many studies on carbon stock assessment have been carried out in *Juniperus* forests across the globe and none have been conducted previously in the juniper forest of Ziarat. In the pinyon-juniper of western Colorado Plateau, mean above-ground woody carbon was estimated to be $5.2 \pm 2.0$ Mg C/ha [13]. In Gilgit Baltistan, Ismail et al. [14] studied the carbon stock of *Juniperus communis* using allometric equation of Jenkin et al. [15] and estimated the amount of carbon to be 1.96 ton/ha. Soils in juniper forests are also considered as a cost-effective carbon sink and conserving this type of forest is imperative for carbon sequestration [16].

The second-largest anthropogenic $CO_2$ emissions are from land-use changes such as rigorous cultivation, deforestation and logging [17]. In the past decades there has been an alarming rise in urbanization, illegal harvesting and agricultural activities [18]. Monitoring land-use and land-cover (LULC) change may help environmentalists, organizations and government offices to devise conservation strategies and management plans [19]. In addition, it could be a significant step to preserve and enhance carbon storage, if the land is properly managed, deforestation is controlled, and reforestation of the degraded land is carried out [20]. Above-ground biomass of forest plays a pivotal role as a terrestrial carbon sink in global carbon cycling [21]. As a carbon sink, forests are an economic treasure worth billions of dollars which currently absorb 30% of all the carbon dioxide emission globally every year [22].

Changes in land use such as converting land for agriculture purposes, results in lowering the soil organic carbon (SOC) or simply depleting carbon in the soil [23]. The conversion of rangeland into agricultural land reduces soil carbon [24]. The world's soil stores significantly much more carbon than the Earth's atmosphere [25]. The loss of soil carbon to the atmosphere may intensify the warming of the planet [26]. The carbon stock in soil has been greatly lost or widely degraded [27]. However, if good management practices are put in place, the SOC levels of the soil may be elevated along with enhancing soil quality [28]. Evidence suggests that SOC is affected by tree species and that some trees may be better at sequestering carbon in the soil [29].

Remote sensing (RS) and geographic information systems (GIS) serve as efficient tools for general and more detailed LULC change analysis [19]. LULC classification is an important research area in remote sensing. It is an established methodology in producing accurate, reliable and updated maps that are significant for ecological monitoring and management [30].

The two main objectives of this study are (1) to find the total loss in forest area of Ziarat over the past three decades using optical satellite imagery (2) to measure the biomass and estimate carbon stock in juniper forest of Ziarat and assess changes in carbon stock.

## 2. Materials and Methods

*2.1. Site Description*

Ziarat district lies in one of the six divisions of Balochistan province and happens to have one of the second-biggest reserves of juniper forest in the world. Juniper forest is unevenly spread between the two tehsils of Ziarat District, which are Sanjavi Tehsil and Ziarat Tehsil. Figure 1 shows the study area map of district Ziarat Balochistan.

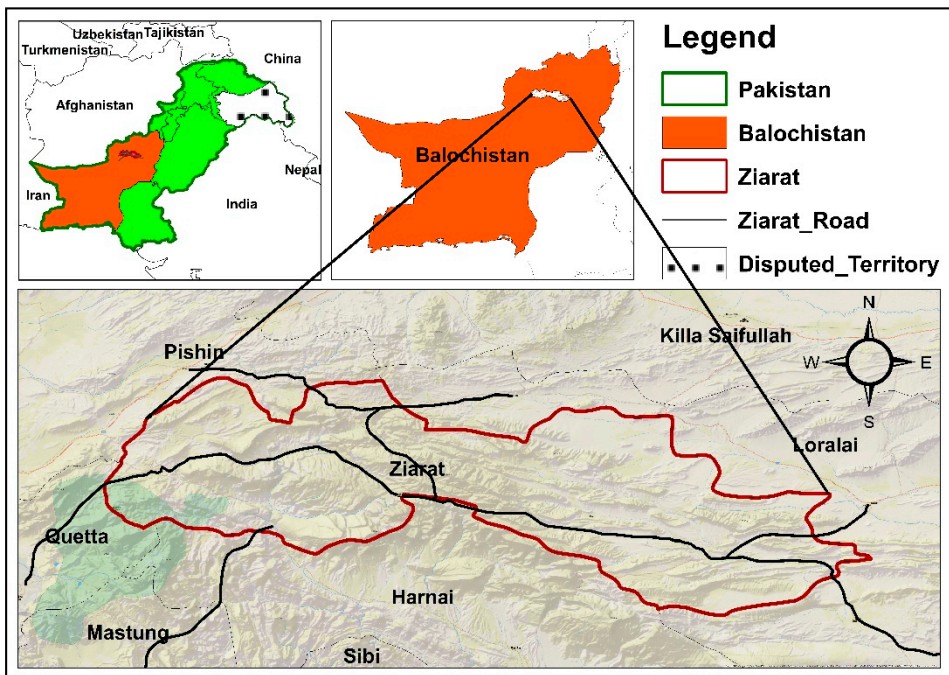

**Figure 1.** Study area showing district Ziarat Balochistan.

### 2.2. Monitoring Land-Use and Land-Cover (LULC) Changes

The archived Landsat imagery provide a unique opportunity to identify land use changes over the past 30 years [31]. Because of that, for the detection of the land cover for District Ziarat, images of Landsat 4-5 TM (Thematic Mapper) and Landsat 8 OLI (Operational Land Imager) with no or minimum cloud cover (0–10%) were selected and downloaded for the years of 1988, 1998, 2008, and 2018 from the USGS (United States Geological Survey) Earth Explorer website (https://earthexplorer.usgs.gov/). Two tiles (path row number 152/39 and 153/39) were downloaded for each period to cover the study area. The downloaded images belong to the month of September (Table 1) since it was found very feasible with almost no clouds present.

**Table 1.** Landsat images used for land cover classification.

| Sensor | Year | Date | Bands (μm) |
|---|---|---|---|
| Landsat 8 OLI | 2018 | 10-September 17-September | Band 2 (0.45–0.51) Band 3 (0.53–0.59) Band 4 (0.64–0.67) Band 5 (0.85–0.88) Band 6 (1.57–1.65) Band 7 (2.11–2.29) |
| Landsat 5 TM | 2008 | 30-September 21-September | Band 1 (0.45–0.52) Band 2 (0.52–0.60) |
|  | 1998 | 03-September 10-September | Band 3 (0.63–0.69) Band 4 (0.77–0.90) |
|  | 1988 | 07-September 14-September | Band 5 (1.55–1.75) Band 7 (2.09–2.35) |

The images were automatically atmospherically corrected using Semi-Automated Classification plugin in QGIS 3.6.2 (QGIS Development Team 2002). Images along with the metadata file were uploaded in the pre-processing Table DOS1 (Dark Object Subtraction) algorithm option was checked okay before running the processing chain. Atmospherically corrected images were opened in Arc-Map 10.3 (Esri 2014, Redlands, CA, USA), where

the bands were stacked using the composite band tab in the Image Analysis window. The stacked images were mosaiced followed by extraction of study area using extract by mask algorithm. The study area was thoroughly examined on Google Earth maps. Both true and false color of Landsat images were observed to ensure that the visual interpretation was done correctly. Once the area was studied, pixels were assigned to the classes and classification was carried out via maximum likelihood classification. The area for each class was calculated in ArcMap and the data were further analyzed by using MS Excel. Line graphs and pie charts were formulated for easy analysis of the data.

*2.3. Carbon Stock Assessment*

This study estimated carbon from all the pools namely above ground tree biomass, shrubs, litter and soil C. Below ground biomass, however, was not considered in this study due to limited resources.

The field work was carried out in the month of February 2018. Six random clusters were selected throughout the study area, each with one primary and four secondary sampling units. The number of total plots were 30, however, due to absence of trees in some areas, the number of plots were reduced to 21 plots as depicted in Figure 2 below.

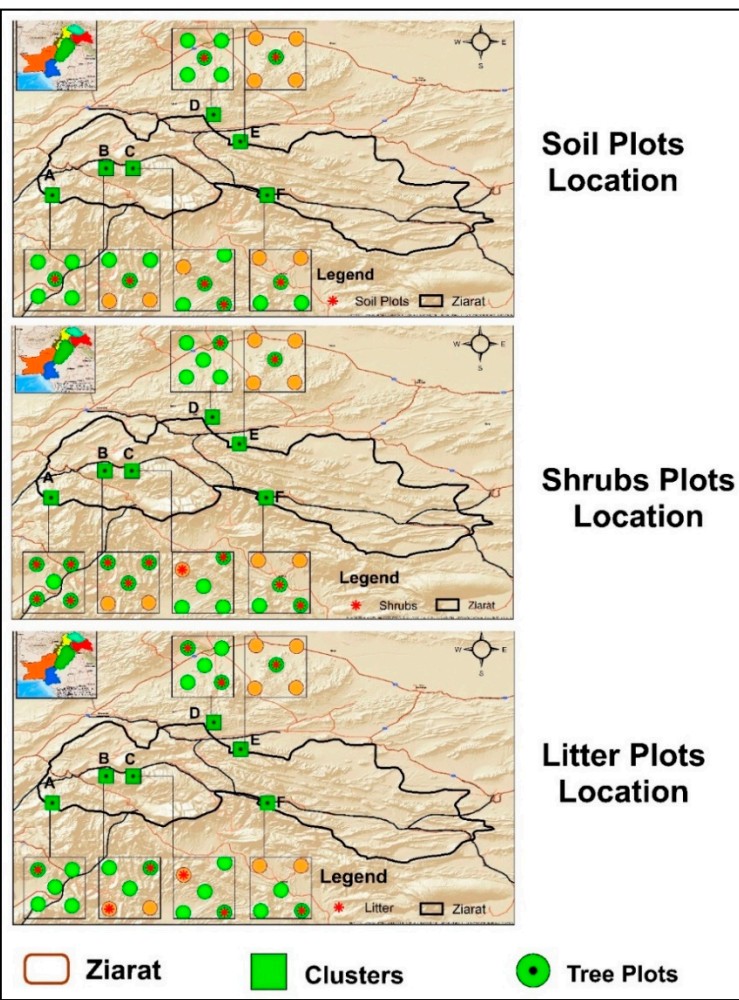

**Figure 2.** Location of tree plots, soil samples, shrubs samples and litter samples in the study area.

The above-ground biomass was calculated for all the plots, however, only one soil sample was collected from the primary plot of each cluster thus giving us six soil samples. Plots for above-ground biomass, shrubs, litter and soil had a radius of 17.8 m, 5.64 m, 0.56 m respectively, keeping in mind that soil and litter plots have the same radius. Shrubs

and litter samples were collected as per their availability, and their location data points are displayed in Figure 2.

Tree height was measured using a clinometer and diameter was measured at breast height of 1.3 m using a caliper. All these data including elevation, site coordinates, time and date of sampling were also recorded. Living biomass data was entered into excel sheets to run the allometric equations. The allometric equations (see Table 2) used for biomass estimation were taken from Jenkin et al. [15], Ali [32] and Chave et al. [33]. The Jenkin et al. [15] equation has been used by the forest department of Gilgit Baltistan [14]. The equation used by Ali [32] and Chave et al. [33] has also been used by WWF (World Wide Fund) for the National Forest Inventory of Pakistan.

**Table 2.** Allometric equation used in the study.

| Jenkin et al. (2003) Equation | Ali. (2015) Equation | Chave et al. (2005) Equation |
|---|---|---|
| $EXP(-0.7152 + 1.7029 \times LN(DBH))$ | $0.1645 \times (p \times D^2 \times H)^{0.8586}$ | $EXP(-2.187 + 0.916 \times LN(WD \times DBH^2 \times H))$ |

*EXP* is the exponential, *LN* is the natural logarithm, *DBH* is the Diameter at Breast Height, *p* is the wood gravity, *D* is the Diameter, *WD* is the Wood Density, and *H* is the Height.

Fraction of carbon in the above ground living biomass can be assessed using the following equation adopted from FAO [34].

$$\text{Total Carbon in above ground biomass (kg)} = 0.475 \times \text{Above ground Biomass (kg)} \tag{1}$$

where the above ground biomass is assessed using the allometric equations listed in Table 2 The above ground living biomass volume formula was derived from the stem biomass formula of FAO [35] as mentioned below:

$$\text{Biomass (kg)} = V_s \times WD \times 1000 \tag{2}$$

where $V_s$ is stem volume and *WD* is wood density. Similarly, tree volume can be calculated by using following equation:

$$\text{Biomass (kg)} = V_t \times WD \times 1000 \tag{3}$$

$$V_t = \text{Biomass}/WD \times 1000 \tag{4}$$

where $V_t$ is tree volume and *WD* is wood density.

The carbon stock of each carbon pool in tons was converted into tons per hectare using the following formula:

$$C \text{ in ton/ha}_{pool} = C \text{ in tons/Area of the plot}_{pool} \tag{5}$$

Total above ground carbon (million tons) for the total forest area, in each respective year of 1988, 1998, 2008 and 2018 was estimated using the following equation:

$$\text{Total forest Above Ground Carbon (million tons)} = C \text{ in ton/ha} \times \text{Area of forest in ha for each year} \tag{6}$$

Total forest carbon for the years before 2018 were estimated using average carbon stock of the three allometric equations used. Additionally, total above carbon for the year 2028, 2038 and 2048 was also estimated using the above formula. Area in hectares of these future years were inferred using the linear forecast model in Microsoft excel worksheet 2013 (Supplementary Material). Total carbon of the forest (million tons) for each year of past and future is presented in results.

The method for estimating total carbon stock of the forest by taking the product of average carbon ha$^{-1}$ and the total forest cover (ha) at a specific temporal period was also used by Mannan et al. [18].

*2.4. Linear Forecast Model*

Linear forecast function uses a linear regression method to predict future values based on historical figures. It is a method of defining the relationship between two or more variables in a way that changes in dependent variable can be accounted by changes in the independent variable. In our study carbon stock and forest area are the dependent variables and time (against year) is the independent variable.

*2.5. Soil Carbon Stocks*

Soil samples were collected at three depths; 0 to10 cm, 10 to 20 cm and 20 to 30 cm. An auger was used for the collection of soil samples. A total of Six soil samples were collected, one from each cluster. Bulk density of each soil sample was calculated in $g/cm^3$. For the determination of carbon in soil, the Walkley–Black titration procedure was applied. To find soil C in grams, the following equation was utilized.

$$SC \text{ (g)} = BD \text{ g/cm}^3 \times SOC \text{ (\%)} \times HT \text{ cm} \times 100 \tag{7}$$

where *SC* Soil Carbon, *BD* is the Bulk Density, *SOC* is the Soil Organic Carbon, and *HT* is the Horizon Thickness.

Soil carbon in grams was converted into tons and further into ton/hectare by dividing it by the plot area.

**3. Results**

*3.1. LULC Changes 1988–2018*

Land cover maps (Figure 3) showed drastic changes in the land use pattern of Ziarat District. Forest area, which is the major focus of the study decreased from 21.5% in 1988 to 15.5% in 2018 showing a decrease of 6% in the total area. This represents significant deforestation over the past decades. Similarly, agriculture area increased from 1.5% in 1988 to 3.5% in 2018. This might be due to an increasing population and demand for agricultural production (Figure 3). As per the housing and population census, the population of Ziarat increased from 32,196 people in 1981 to 160,422 people in 2017. Barren land also significantly increased from 76.5% in 1988 to 81% in 2018 due to deforestation.

As shown in Figure 4 (land use change trend and forecast), the forest area decreased from 71,005 hectares in 1988 to 50,311 hectares in 2018, depicting a loss through deforestation of 20,694 hectares in the past 30 years. On the other hand, the agriculture land increased from 4848 hectares in 1988 to 11,625 hectares in 2018 representing an increase of 6777 hectares.

Figure 4 also shows the linear forecast values inferred for next 30 years. The forecast analysis shows that the forest area will decrease to 29,976.5 hectares in 2048, while the agricultural and the barren land area will increase to 17,826.7 hectares and 279,259 hectares, respectively, in the coming three decades.

Table 3 below shows the accuracy assessment of the land-use maps of Ziarat district. Accuracy assessment was performed in Arc Map using the confusion matrix, where 140 training points were selected for each temporal map. The confusion matrix shows producer's and user's accuracy for each land-use category of the respective yearly maps. Accuracy from the perspective of the map designer is termed as the producer's accuracy and it shows the probability of whether land cover has been classified correctly. User's accuracy tells us the correctness of the map from a user's viewpoint. It tells us the probability of how often the classified class on the map will be present on the ground [36,37]. It is quite high and above 90%. The overall accuracy of the classification for 1988, 1998, 2008 and 2018 was 0.97, 0.98, 0.98, and 0.97. Kappa analysis was undertaken if the performance of the classification did well in comparison to randomly assigning values. This ranged from −1 to 1 and a value of zero represents that the classification was no better random value assignment [36,37]. The Kappa coefficient was 0.96, 0.98, 0.97 and 0.96, respectively, for each consecutive year.

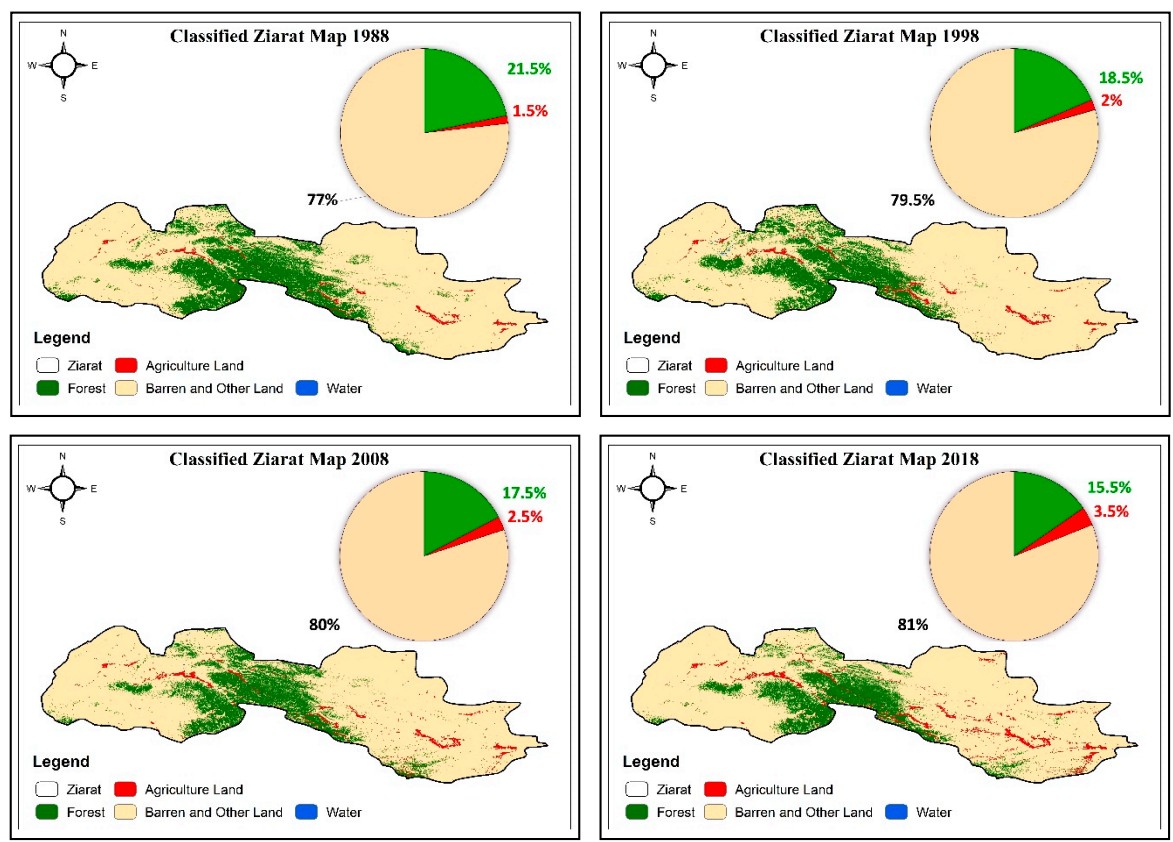

**Figure 3.** Land-use and land-cover (LULC) maps of the study area for the year 1988, 1998, 2008 and 2018.

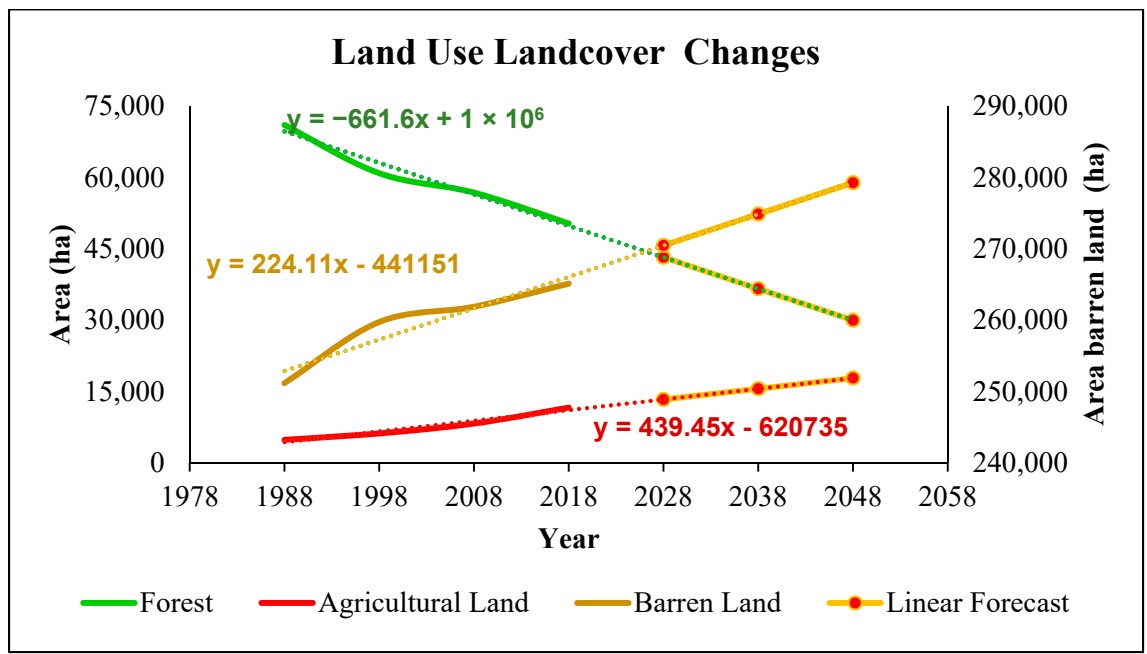

**Figure 4.** Land-use change trend (1988–2018) with inferred values for future 30 years using linear forecast.

**Table 3.** Accuracy assessment of land use maps.

| | 1988 | | 1998 | | 2008 | | 2018 | |
|---|---|---|---|---|---|---|---|---|
| | Producer's Accuracy % | User's Accuracy % | Producer's Accuracy% | User's Accuracy% | Producer's Accuracy % | User's Accuracy % | Producer's Accuracy % | User's Accuracy % |
| Forest | 92 | 100 | 100 | 100 | 94 | 100 | 92 | 100 |
| Agriculture | 100 | 100 | 100 | 100 | 100 | 100 | 100 | 100 |
| Barren and Other land | 100 | 92.59 | 100 | 100 | 100 | 94.33 | 100 | 92.59 |
| Water | 100 | 100 | 80 | 100 | 100 | 100 | 100 | 100 |
| Kappa Coefficient | 0.96 | | 0.98 | | 0.97 | | 0.96 | |
| Overall Accuracy | 0.97 | | 0.98 | | 0.98 | | 0.97 | |

*3.2. Carbon Stock Assessment*

Table 4 shows plot-wise estimates of different parameters of the juniper tree namely the number of trees, average height, average diameter and basal area. The maximum number of trees was recorded in plot no 14 which is 58 trees, and the minimum number of trees was in plot 21 with 10 individuals. The maximum average height and diameter were in plot 18 and the minimum average height and diameter were in plot 15. The maximum and minimum average height is 5.81 and 4.38 m, respectively. The highest and lowest diameter came out to be 23.94 and 10.01 cm. The maximum basal area was in plot 17 of 2.25 m$^2$. The minimum basal area was 0.19 m$^2$ in plot 21.

**Table 4.** Parameter estimation of sampled plots.

| Plot ID | Number of Trees | Average Height (m) | Average Diameter (cm) | Basal Area (m$^2$) |
|---|---|---|---|---|
| Plot 1 | 14 | 4.96 | 18.27 | 0.72 |
| Plot 2 | 20 | 4.65 | 13.71 | 0.48 |
| Plot 3 | 30 | 5.01 | 17.33 | 1.36 |
| Plot 4 | 22 | 4.60 | 11.22 | 0.25 |
| Plot 5 | 26 | 4.84 | 12.80 | 0.40 |
| Plot 6 | 24 | 5.26 | 17.91 | 0.85 |
| Plot 7 | 20 | 4.62 | 12.43 | 0.36 |
| Plot 8 | 24 | 5.16 | 16.55 | 0.67 |
| Plot 9 | 46 | 4.64 | 12.50 | 0.86 |
| Plot 10 | 23 | 5.13 | 18.43 | 0.97 |
| Plot 11 | 16 | 4.75 | 12.29 | 0.22 |
| Plot 12 | 21 | 4.79 | 12.84 | 0.33 |
| Plot 13 | 38 | 4.89 | 14.36 | 0.87 |
| Plot 14 | **58** | 4.72 | 13.27 | 1.29 |
| Plot 15 | 55 | **4.38** | **10.01** | 0.55 |
| Plot 16 | 21 | 4.44 | 11.83 | 0.36 |
| Plot 17 | 50 | 5.17 | 18.35 | **2.25** |
| Plot 18 | 16 | **5.81** | **23.94** | 0.98 |
| Plot 19 | 21 | 5.61 | 20.66 | 0.86 |
| Plot 20 | 35 | 4.98 | 16.73 | 1.23 |
| Plot 21 | **10** | 4.95 | 14.2 | **0.19** |
| | | **4.92** | **15.22** | **0.76** |

The bold in this graph depicts the maximum and minimum values in all the plots.

The overall average height of all the plots was 4.92 m while the overall average diameter was 15.22 cm. The overall average basal area was 0.76 m$^2$.

Table 5 below shows the total biomass and the total volume of juniper forest using three different above-ground allometric equations. Total biomass calculated from all three equations were 37,281.51 kg, 24,812.12 kg and 37,518.67 kg respectively. The volume

estimated using these equations were 74.56 m$^3$, 69.62 m$^3$ and 75.04 m$^3$. It can be noted that there was a small difference between the values of the equations of Jenkin et al. [15] and Chave et al. [33]. But there was a visible difference in the biomass and volume calculated using the equation of Ali [32].

**Table 5.** Comparison of total biomass and total volume of the three respective allometric equations.

| | Jenkin et al. (2003) | | Ali (2015) | | Chave et al. (2005) | |
|---|---|---|---|---|---|---|
| Serial no | Total Biomass kg | Total Volume m$^3$ | Total Biomass kg | Total Volume m$^3$ | Total Biomass kg | Total Volume m$^3$ |
| Plot 1 | 1469.48 | 2.94 | 1525.63 | 3.05 | 1733.31 | 3.47 |
| Plot 2 | 1136.97 | 2.27 | 1042.32 | 2.08 | 1108.32 | 2.22 |
| Plot 3 | 2808.46 | 5.62 | 2882.99 | 5.77 | 3273.74 | 6.55 |
| Plot 4 | 726.67 | 1.45 | 564.27 | 1.13 | 551.60 | 1.10 |
| Plot 5 | 1092.21 | 2.18 | 894.63 | 1.79 | 898.08 | 1.80 |
| Plot 6 | 1969.00 | 3.94 | 1857.40 | 3.71 | 1998.63 | 4.00 |
| Plot 7 | 900.20 | 1.80 | 779.92 | 1.56 | 807.14 | 1.61 |
| Plot 8 | 1637.96 | 3.28 | 1471.67 | 2.94 | 1539.76 | 3.08 |
| Plot 9 | 2121.05 | 4.24 | 1868.06 | 3.74 | 1959.28 | 3.92 |
| Plot 10 | 2148.44 | 4.30 | 2109.62 | 4.22 | 2317.58 | 4.64 |
| Plot 11 | 589.45 | 1.18 | 476.99 | 0.95 | 475.87 | 0.95 |
| Plot 12 | 889.08 | 1.78 | 725.82 | 1.45 | 725.00 | 1.45 |
| Plot 13 | 2139.73 | 4.28 | 1907.67 | 3.82 | 1999.62 | 4.00 |
| Plot 14 | 3057.40 | 6.11 | 2791.66 | 5.58 | 2988.37 | 5.98 |
| Plot 15 | 1567.66 | 3.14 | 1217.62 | 2.44 | 1200.58 | 2.40 |
| Plot 16 | 826.38 | 1.65 | 778.57 | 1.56 | 851.04 | 1.70 |
| Plot 17 | 4782.44 | 9.56 | 4805.25 | 9.61 | 5381.93 | 10.76 |
| Plot 18 | 2107.65 | 4.22 | 2115.15 | 4.23 | 2346.73 | 4.69 |
| Plot 19 | 2021.00 | 4.04 | 1893.39 | 3.79 | 2019.08 | 4.04 |
| Plot 20 | 2786.63 | 5.57 | 2679.93 | 5.36 | 2914.52 | 5.83 |
| Plot 21 | 503.65 | 1.01 | 423.55 | 0.85 | 428.50 | 0.86 |
| Total | **37,281.51** | **74.56** | **34,812.12** | **69.62** | **37,518.67** | **75.04** |

Figure 5 represents the correlation between the parameters of tree as specified in Tables 3 and 4. There is a good correlation between the height and diameter I-e 0.89, and this does not vary among the allometric equation. This shows that the height of the tree increases with the increase in diameter of the juniper tree.

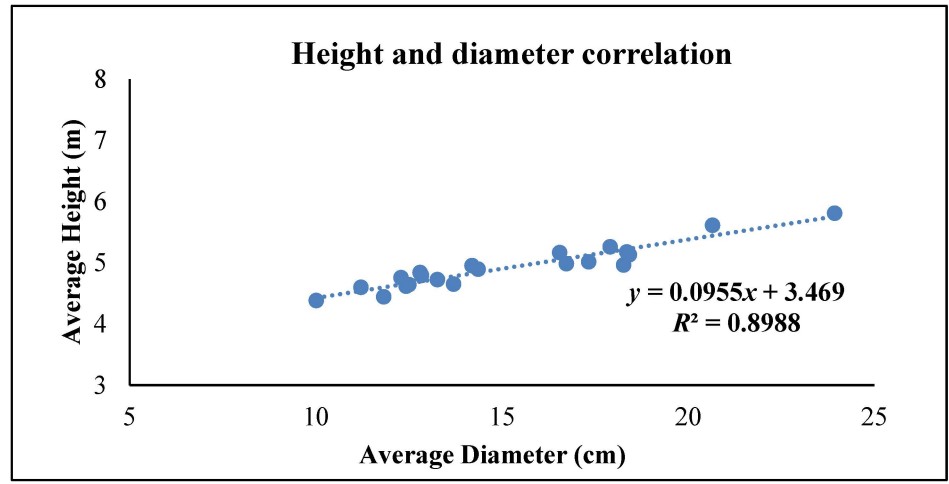

**Figure 5.** Correlation of average height and average diameter from the sampled plots.

Table 6 shows an overall plot summary of carbon stock in above ground pool using the three equations as mentioned earlier. Plot 17 contains the highest amount of above ground biomass i.e., 22.48, 22.58 and 25.30 ton/ha, and the lowest above ground biomass is found in plot 21 i.e., 2.37, 1.99, and 2.01 ton/ha. The overall average living biomass of the forest of all plots using the three equations is 8.34, 7.79 and 8.4 ton/ha, respectively.

**Table 6.** Comparison of the above ground carbon using the allometric equations from Jenkin et al. [15], Ali [32], and Chave et al. [33].

| Serial. No | ABG (t/ha) Using Jenkin et al. (2003) Allometric Equation | ABG (t/ha) Using Ali (2015) Allometric Equation | ABG (t/ha) Using Chave et al. (2005) Allometric Equation |
|---|---|---|---|
| Plot 1 | 6.91 | 7.17 | 8.15 |
| Plot 2 | 5.34 | 4.90 | 5.21 |
| Plot 3 | 13.20 | 13.55 | 15.39 |
| Plot 4 | 3.42 | 2.65 | 2.59 |
| Plot 5 | 5.13 | 4.20 | 4.22 |
| Plot 6 | 9.25 | 8.73 | 9.39 |
| Plot 7 | 4.23 | 3.67 | 3.79 |
| Plot 8 | 7.70 | 6.92 | 7.24 |
| Plot 9 | 9.97 | 8.78 | 9.21 |
| Plot 10 | 10.10 | 9.92 | 10.89 |
| Plot 11 | 2.77 | 2.24 | 2.24 |
| Plot 12 | 4.18 | 3.41 | 3.41 |
| Plot 13 | 10.06 | 8.97 | 9.40 |
| Plot 14 | 14.37 | 13.12 | 14.05 |
| Plot 15 | 7.37 | 5.72 | 5.64 |
| Plot 16 | 3.88 | 3.66 | 4.00 |
| Plot 17 | 22.48 | 22.58 | 25.30 |
| Plot 18 | 9.91 | 9.94 | 11.03 |
| Plot 19 | 9.50 | 8.90 | 9.49 |
| Plot 20 | 13.10 | 12.60 | 13.70 |
| Plot 21 | 2.37 | 1.99 | 2.01 |
| Average | **8.34** | **7.79** | **8.40** |

Table 7 below shows the carbon estimation of the pools: soil, shrubs and litter. The respective plots where there is highest and lowest carbon stock have been highlighted. The overall average soil carbon of the juniper forest is 24.35 ton/ha. The overall carbon stored in shrubs and litter is 4.66 and 1.52 ton/ha, respectively.

Figure 6 below shows the total biomass of the juniper forest of Ziarat since 1988 using three different equations. The initial bars on the left of the graph show the total estimated carbon stock of the forest for the years 1988, 1998, 2008 and 2018. The bars on the right represent the inferred values of the total forest carbon stock for the year 2028, 2038 and 2048.

The Jenkin et al. [15] equation in blue bars estimates total above ground carbon stock of the forest in 1988 as 0.59 million tons. It reduced to 0.51 million tons, 0.47 million tons and 0.42 million tons in 1998, 2008, and 2018, respectively. The forecast values using the same equation shows that the total carbon of the forest may reduce to 0.36 million tons in 2028, 0.31 million tons in 2038 and 0.25 million tons in 2048.

Similarly, according to the Ali [32] allometric equation estimates, presented in the orange bar (Figure 6), the total above ground carbon stock of the forest in 1988 was 0.55 million tons. It reduced to 0.47 million tons, 0.44 million tons, and 0.39 million tons in 1998, 2008 and 2018, respectively. The forecast values show that the total carbon of the juniper forest may reduce to 0.34 million tons in 2028, 0.29 million tons in 2038 and 0.23 million tons in 2048.

Moreover, the estimates of the Chave et al. [33] allometric equation, displayed as green bars, are also interesting and almost like the Jenkin et al. 2003 allometric equation. (Figure 6). The total above ground carbon stock of the forest in 1988 was 0.60 million tons. It reduced to 0.51 million tons, 0.48 million tons and 0.42 million tons in 1998, 2008 and 2018, respectively. The forecast values show that the total carbon of the juniper forest may reduce to 0.36 million tons in 2028, 0.31 million tons in 2038 and 0.25 million tons in 2048.

**Table 7.** Estimated carbon in soil, shrubs and litter in sampled plots.

| Serial. No | Soil Carbon ton/ha | Shrubs ton/ha | Litter ton/ha |
|---|---|---|---|
| Plot 1 | | 0.057 | |
| Plot 2 | | 0.045 | 2.22 |
| Plot 3 | **28.06** | 0.033 | |
| Plot 4 | | 0.042 | |
| Plot 5 | | | |
| Plot 6 | | 0.033 | **3.83** |
| Plot 7 | 27.22 | 0.075 | |
| Plot 8 | | 0.016 | 1.87 |
| Plot 9 | | | |
| Plot 10 | 27.54 | 0.085 | **1.37** |
| Plot 11 | | **0.102** | |
| Plot 12 | | | 1.23 |
| Plot 13 | | | |
| Plot 14 | | | 0.55 |
| Plot 15 | **17.76** | 0.066 | |
| Plot 16 | | | 0.45 |
| Plot 17 | | 0.047 | |
| Plot 18 | 22.53 | **0.014** | 0.55 |
| Plot 19 | | | |
| Plot 20 | 22.99 | 0.019 | 1.62 |
| Plot 21 | | 0.018 | |
| Average | **24.35** | **0.05** | **1.52** |

The bold values are the high and the low values in all the data set.

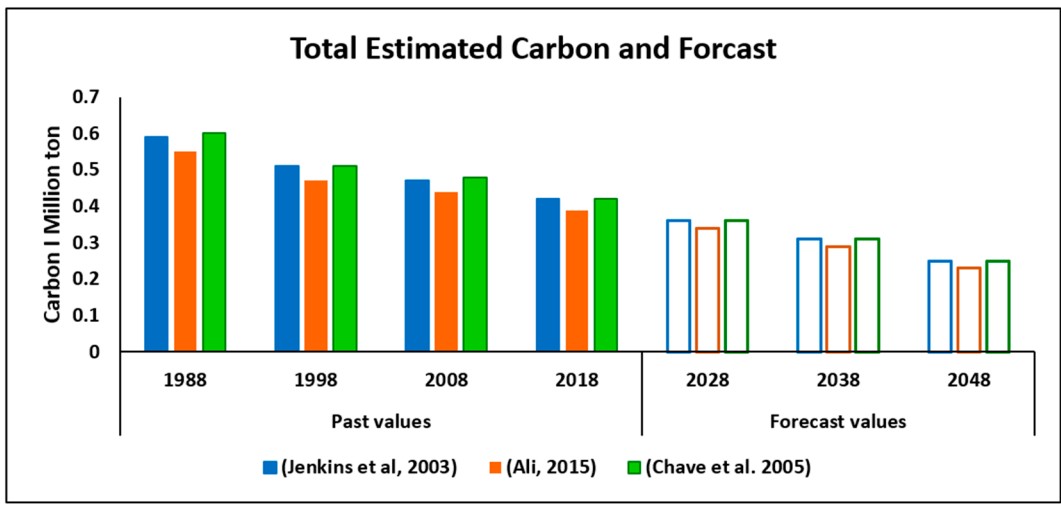

**Figure 6.** Total forest carbon in million tons using the equations of Jenkin et al. 2003, Ali (2015) and Chave et al., 2005 from 1988 to 2018 and for the future 30 years using linear forecast analysis.

## 4. Discussion

Ziarat juniper forest was declared a biosphere reserve in 2013. Such forests will play a pivotal role in future in furthering the cause of carbon sequestration [10]. However, they face a visible threat of deforestation and forest degradation [38]. Major drivers of deforestation and forest degradation are population expansion, agriculture intensification, fuel wood consumption, poor regeneration, illegal cutting, overgrazing, canopy dieback, mistletoe attack, periodic drought and medicinal use of the juniper tree [38,39]. For identifying deforestation, a quantitative assessment of the land-use change was performed for four land use classes namely, forest, agriculture, water and barren land. Focus of the study was reduction in forest cover, which according to our results was accounted for by the increase in agricultural area.

There is an inverse relationship between forest resources and both population as well as the amount of land used for agriculture. With population increase, more and more forests are encroached upon for harvesting firewood and timber, thereby increasing the non-forest area in Ziarat district [40]. The population increase has led to the increase in dependency on agriculture [12], thus having a synergetic effect on the forest. Besides other drivers, *Archeothobium oxycedri*, known as dwarf mistletoe, also damages the *Juniperus* species [41]. Sarangzai et al. [42] also reported the spread of the same parasitic and infectious plant in the juniper forest of Ziarat. Harvesting fuelwood is also one of the major drivers of deforestation since there is no other source of fuel to keep the local people warm in winter [43]. Even though natural gas has been provided to Ziarat, the pressure in gas pipelines is extremely low thus compelling residents to cut trees in the winter season [44]. Most of the juniper forest have low seedlings and show no adequate regeneration thus slowing down recovery time [45].

Urban areas were not classified, because most of the construction sites are either muddy houses or wooden-built sites, thus making it extremely difficult to separate its spectral information in Landsat imagery. The area also has myriad soil forms and colors resulting in the overlapping of the pixels during classification. Even the forest area was difficult to classify since the juniper forest is a sparse forest with less tree density. However, it was classified very well when observed simultaneously with Google Earth imagery. The results of this study are also comparable to the results of a study conducted by WWF and SUPPARCO (Pakistan Space & Upper Atmosphere Research Commission) in 2012 on the Juniper Forest of Ziarat using SPOT data [11]. This suggests that the Landsat data are good for forest classification and land-use change.

Ismail et al. [14] estimated the above ground carbon per hectare of *Juniperus communis*. The total number of juniper trees counted were 278 and the estimated average carbon was 1.96 ton/ha with a total basal area of 12.28 $m^2$. This value is substantially less compared to our estimated carbon of 8.34 ton/ha and a total basal area of 16.06 $m^2$ using the same equation for *Juniperus excelsa*.

Since no specific equations exist for the *Juniperus excelsa* species, it is recommended that the forest department in collaboration with academia develop an allometric equation for it. If developing an equation is not possible or not within the capacity, it is recommended that the three equations used in the study are used for any future carbon stock study. The three equations used in this study gave very similar results.

The data were collected from 21 plots having 585 trees. The results show that the soil contains more carbon than trees, which may be attributed to the compact soil of the area, low temperature conditions and the less dense/sparse nature of the forest. It may also be due to the age of the forest, which is very old thus, accumulating humus for thousands of years. One of the major reasons for less biomass in the tree is small height and the very sparse nature of the forest reserve.

The juniper forest of Ziarat is an extremely rare forest and needs to be protected and sustained. The forest area has decreased from 10,025 hectares in 1960 to 53,092 hectares in 2010 and is a clear manifestation of threats faced by the juniper forest of Ziarat, Balochistan. [5,11]. This comprises multiple factors. The first one is the population of the area that has dramatically increased (about 400%) from 1981 to 2017 as per the population and housing census. Besides marble mining at few places, there is no such industrial activity in the Ziarat district. Therefore, the population must depend on natural resources for their daily subsistence. The agriculture area has also increased (2% from 1988 to 2018) as indicated in the previous section of this study. Other factors include medicinal use of berries, climate change, timber extraction, tourism and poor forest management by the forest department. All these factors are clearly posing an enhanced threat to preservation of these ancient forests of Ziarat. Keeping the current scenario and past practices in view, it is most probable that the forest area will keep on shrinking as mentioned by various studies presented in Figure 6.

Moreover, the protection regime should not be only limited to Ziarat but also to the juniper forest resources in Loralai, Harnai, Quetta and Pishin. Despite all the services the juniper forest provides, it has not been taken care of in a viable way resulting in its deforestation and degradation. If a similar trend of disregard for this precious and ancient forest continues, we might only study about this archeological heritage in archives. For this purpose, the forest department must take practical steps for the conservation of the juniper forest of Balochistan.

## 5. Conclusions

The study concluded that the ancient juniper forest of Ziarat is an important carbon sink, storing a significant amount of carbon in all its pools i.e., above-ground live tree biomass, soil, shrubs and litter. The soil of the juniper forest stores more carbon than the living biomass due to low canopy cover and scattered growth of the trees. Similarly, the litter contains more carbon stock than the shrubs since the quantity of litter found in the plots was higher than the shrubs. Land-use maps showed a tremendous change in the forest cover of Ziarat Balochistan from 1988 to 2018 with a decreasing forest trend and increasing agricultural trend. Furthermore, carbon stock of juniper has also decreased over the past three decades due to excessive deforestation and forest degradation. Clearly, the juniper forest is threatened and if the same pace of deforestation continues, it is very likely that this forest will soon be wiped out. Additionally, the carbon stock of the juniper forest was assessed using three different equations which gave similar results, so it is suggested that these equations may be used for future carbon stock studies on the juniper forest. It is also concluded that, if no appropriate steps are taken towards the conservation of this forest, we may lose the ancient world biosphere reserve in a very short time.

**Supplementary Materials:** The following are available online at https://www.mdpi.com/1999-4907/12/1/51/s1, Table S1: Linear forecast model.

**Author Contributions:** Conceptualization, H.J., M.F.K. and W.R.K.; Methodology, H.J. and N.u.S.; Formal Analysis, K.A.K. and H.J; Data, H.J.; Writing Original Draft Preparation, H.J., W.R.K. and M.F.K.; Writing Review & Editing, M.N.; Visualization, U.T.; Project Administration, M.F.K.; Funding Acquisition, M.N. All authors have read and agreed to the published version of the manuscript.

**Funding:** The authors are grateful to Universiti Putra Malaysia (UPM) and NUST for providing financial support from the postgraduate RnD fund from S and T Mega Project to conduct this study.

**Acknowledgments:** The authors acknowledge all the members of research group (C-CARGO) and Universiti Putra Malaysia researchers for being supportive.

**Conflicts of Interest:** The authors declare no conflict of interest.

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
