# Peer review of "Monitoring Carbon Stock and Land-Use Change in 5000-Year-Old Juniper Forest Stand of Ziarat, Balochistan, through a Synergistic Approach"

_forests, doi:10.3390/f12010051_

Round 1

Reviewer 1 Report

I found the paper easy to understand and while not very original in terms of its methods or conclusions, still useful in light of the lack of studies in these forests and the usefulness of the information on land use change to policymakers. I think the authors should try to add some additional information and discussion about past and future population trends, pressures on land for resources, agriculture, and the likelihood of being able to mitigate or preserve the juniper forest through changes in agricultural practices, technology, legislation, or access to resources.

The content of the paper is easily understood, although it is obvious that the authors are not native English speakers. It would take a native English speaker a couple of hours to make minor wording changes that would improve the appearance of the paper. 

Specific suggestions and corrections:

Line 38: Earth's should start with an upper case letter.

Line 48-52: The authors need to state in the text (not leaving it for the reader to check the references) if these estimates of land cover change are separated by years of time, and therefore represent a change in area and not just a difference in the estimated size, caused perhaps by different assessment methodologies.

Line 54: Change "was done" to "have been conducted previously".

Line 68: Instead of "a huge amount" the authors could cite a number or range for the estimated annual forest carbon sink.

Line 75: The phrase "conservation of the land" is too general. The authors are clearly concerned with conservation or preservation of the juniper forests.

I find the legend in Figure 1 difficult to interpret.

Figures 2-4 could probably be combined if a more sophisticated set of markers is employed.

Line 138: WWF needs to be defined.

Table 2: Some of the symbols in the equations are not defined in the text.

Line 165: Change "in 9, figure 10 and figure 11" to "in Figures 9-11".

Lines 170-173 and 206-208: Do the authors know of contributing factors to any changes or trends in the plot-level carbon stocks? If so, is it reasonable to assume that such changes will continue, and that the response of the forest will be the same? i.e. Is the population projected to continue to increase? Are other suitable lands available for agriculture? Are prohibitions on cutting the forests enforceable? Are other fuel sources for winter heating expected to become more available and reliable?

Line 176: Change "Soil sample was collected" to "Soil samples were collected".

Lines 191-195: Are the percentages known to three decimal places with confidence?

Figure 7: Do the linear forecasts preserve the total land area over time?

Line 212: I had to define producer's accuracy and users' accuracy when I published using these terms.

Line 214: Define the Kappa coefficient.

Section 3.2: I am not opposed to listing maximum and minimum numbers of trees or basal area in plots, but it would be more useful for readers and for comparison with other forests if numbers could also be expressed as stems per hectare and the basal area in m² per hectare.

Figures 9, 10 and 11 could easily be combined into one with multiple bars, and this would allow some short paragraphs to be combined, space to be saved, and comparisons to be much easier to see (i.e. line 291-292).

Line 304: Change "land use changes were performed" to "a quantitative assessment of land use change was performed".

Line 307: Change "agriculture/population" to "both population as well as the amount of land used for agriculture".

Line 312: I don't think it reads well to begin a sentence with a citation number. I would use the author's name.

Line 312: The dwarf mistletoe is a plant. The term disease has not been explained.

Line 316: I don't know what is meant be "and show no adequate".

Line 328: Basal area should be expressed as m² per hectare.

Line 335: Change "the data was collected" to "the data were collected".

Author Response

Response to reviewer’s comment on “Monitoring Carbon Stock and Land Use Change in 5,000-year-Old Juniper Forest Stand of Ziarat, Balochistan, through a Synergistic Approach”

The authors are grateful to the anonymous reviewers and editor for their positive assessment and the constructive comments. Efforts are made to address most of their comments and recommendations to further improve the quality of work presented in this manuscript.

Following the recommendations/comments of reviewers, authors made some major changes (in red) in the manuscript as given below:

Anonymous reviewer # 1

I found the paper easy to understand and while not very original in terms of its methods or conclusions, still useful in light of the lack of studies in these forests and the usefulness of the information on land use change to policymakers. I think the authors should try to add some additional information and discussion about past and future population trends, pressures on land for resources, agriculture, and the likelihood of being able to mitigate or preserve the juniper forest through changes in agricultural practices, technology, legislation, or access to resources.

The authors have made the necessary changes and additions in the discussion section relating to the areas mentioned by the reviewer. The following paragraph has been added:

“Juniper forest of Ziarat is an extremely rare forest and needs to be protected and sustained. The forest area has decreased from 10,025 hectares in 1960 to 53,092 hectares in 2010 and is a clear manifestation of threats faced by the Juniper Forest of Ziarat, Balochistan. [2,8]. It is contributed by multiple factors. The first one is the population of the area that has dramatically increased (about 400%) from 1981 to 2017 as per the population and housing census. Besides of marble mining at few places, there is no such industrial activity in the Ziarat district. Therefore, population must depend on natural resources for their daily subsistence. The agriculture area has also increased (2% from 1988 to 2018) as indicated in the previous section of this study. Other factors include medicinal use of berries, climate change, timber extraction, tourism and poor forest management of forest department. All these factors are clearly posing an enhanced threat to preservation of these ancient forests of Ziarat. Keeping the current scenario and past practices in view, it is most probable that the forest area will keep on shrinking as mentioned by various studies presented in Figure 6”.

The content of the paper is easily understood, although it is obvious that the authors are not native English speakers. It would take a native English speaker a couple of hours to make minor wording changes that would improve the appearance of the paper. 

The paper has been reviewed thoroughly and more efforts are put to improve the grammatical errors, language and typos.

Specific suggestions and corrections:

Line 38: Earth's should start with an upper-case letter.

Done

Line 48-52: The authors need to state in the text (not leaving it for the reader to check the references) if these estimates of land cover change are separated by years of time, and therefore represent a change in area and not just a difference in the estimated size, caused perhaps by different assessment methodologies.

Done. The changes have been made and the following sentence has been incorporated:

“The area of this Juniper forest as per the working plan 1960 is 247,166 acres (100025 ha)”

Line 54: Change "was done" to "have been conducted previously".

Done

Line 68: Instead of "a huge amount" the authors could cite a number or range for the estimated annual forest carbon sink.

Done

Line 75: The phrase "conservation of the land" is too general. The authors are clearly concerned with conservation or preservation of the juniper forests.

The sentence has been rephrased to improve the clarity as under:

Carbon stock in soil has been greatly lost or widely degraded [24]. However, if good management practices are put in place, the SOC levels of the soil may be elevated along with enhancing soil quality

I find the legend in Figure 1 difficult to interpret.

The legend has been edited and the Figure has been made more vividly presented.

Figures 2-4 could probably be combined if a more sophisticated set of markers is employed.

The authors have combined the Figures 2-4 in one Figure.

Line 138: WWF needs to be defined.

Done

Table 2: Some of the symbols in the equations are not defined in the text.

Done

Line 165: Change "in 9, figure 10 and figure 11" to "in Figures 9-11".

Done

Lines 170-173 and 206-208: Do the authors know of contributing factors to any changes or trends in the plot-level carbon stocks? If so, is it reasonable to assume that such changes will continue, and that the response of the forest will be the same? i.e. Is the population projected to continue to increase? Are other suitable lands available for agriculture? Are prohibitions on cutting the forests enforceable? Are other fuel sources for winter heating expected to become more available and reliable?

The authors are aware of the contributing changes in the current business as usual scenario, which are affecting the Juniper forest of Ziarat. These contributing factors include the agriculture, population and enforcement etc. The authors work on these areas is pending due to COVID-19.

Line 176: Change "Soil sample was collected" to "Soil samples were collected".

Done

Lines 191-195: Are the percentages known to three decimal places with confidence?

Calculated land use percentage are subject to landcover classification accuracy (Table 3). Percentage with three decimal places are not with confidence, therefore they have been changed to round figure

Figure 7: Do the linear forecasts preserve the total land area over time?

These forecasts are actually representing the business as usual scenario without any extra measures to preserve the forest.  

Line 212: I had to define producer's accuracy and users' accuracy when I published using these terms.

Both the user’s and producer’s accuracy have been defined.

Line 214: Define the Kappa coefficient.

Done

Section 3.2: I am not opposed to listing maximum and minimum numbers of trees or basal area in plots, but it would be more useful for readers and for comparison with other forests if numbers could also be expressed as stems per hectare and the basal area in m² per hectare.

The Juniper forest of Ziarat is unevenly distributed and it can be seen from the number of trees per plot ranging from 14 to 58. Although it is a good recommendation to express stem per hectare and basal area in m² per hectare, however this will create ambiguity in the data since our plots are approximately 0.1 hectare, therefore, it would not be good if we randomly estimate the mentioned factors per hectare.

Figures 9, 10 and 11 could easily be combined into one with multiple bars, and this would allow some short paragraphs to be combined, space to be saved, and comparisons to be much easier to see (i.e. line 291-292).

The authors have combined Figure 9 to 11.

Line 304: Change "land use changes were performed" to "a quantitative assessment of land use change was performed".

Done

Line 307: Change "agriculture/population" to "both population as well as the amount of land used for agriculture".

Done

Line 312: I don't think it reads well to begin a sentence with a citation number. I would use the author's name.

The citation number has been replaced with the author’s name.

Line 312: The dwarf mistletoe is a plant. The term disease has not been explained.

The term “disease” has been changed with “parasitic and infectious plant”.

Line 316: I don't know what is meant be "and show no adequate".

The sentence has been rephrased as under:

“Most of the juniper forest have low seedlings and show no adequate regeneration thus slowing down recovery time”.

Line 328: Basal area should be expressed as m² per hectare.

The value presented on these lines are the total basal area of all the plots combined. We cannot change this into basal area in m² per hectare because referred paper has studied many species in 556 plots and where Juniperus is uncommon. However, our study has only 21 plots where the only focus is Juniperus. If we convert them into per hectare value using the area sampled, there will be great vagueness. Therefore, we suggest that same total basal area value be used for comparison.

Line 335: Change "the data was collected" to "the data were collected".

Done

Reviewer 2 Report

The manuscript provides an interesting results accessing forest cover change and carbon stock of ancient Juniper Forest in Ziarat. A few points (see below) need clarification or language editing.

L36: You refer to FAO FRA 2015, while FRA 2020 already published and available.

L58: “an ideal and active carbon sink” What do you mean?

L83: please use “estimate” instead of “calculate”

L111: Please clarify what does it mean: “The map data was extracted to excel to assess the study area in hectares”.

L114: “living biomass, shrubs” – shrubs are also “living biomass”, do you mean “above-ground live biomass of trees, shrubs”?

L118: “non-availability of trees”, do you mean “absence”?

L130: “which is why”

L143, 147: The equations do not consider that above-ground biomass includes branches, while stem volume not.

L177: “Five samples were collected from different clusters”. It is not clear if five sample collected from every sample plot or five sample totally for the region.

L211: “training points” – did you use for the validation same points than were used for training the maximum likelihood classification algorithm?

L221: “plot no 14”

L224: “10.01 cm” One cannot measure tree diameter with such precision

Table 4: “Sr. No; No of Trees” – do you mean “plot ID” and “number of trees”?

Table 4: “Average Diameter (cm); Basal Area m2” Please put units (m2) uniformly in brackets

Figure 9-11: There is no sense for me to make tree separate predictions, one (mean) would be enough.

L312: “Felling trees for burning purposes” – Harvesting fuel wood?

L328: “very less”, maybe “substantially less” or “much less”

L350: “above ground, …, shrubs”. Do you mean “above-ground live tree biomass, …, shrubs”

Author Response

Response to reviewer’s comment on “Monitoring Carbon Stock and Land Use Change in 5,000-year-Old Juniper Forest Stand of Ziarat, Balochistan, through a Synergistic Approach”

The authors are grateful to the anonymous reviewers and editor for their positive assessment and the constructive comments. Efforts are made to address most of their comments and recommendations to further improve the quality of work presented in this manuscript.

Following the recommendations/comments of reviewers, authors made some major changes (in red) in the manuscript as given below:

Anonymous reviewer # 2

The manuscript provides an interesting result accessing forest cover change and carbon stock of ancient Juniper Forest in Ziarat. A few points (see below) need clarification or language editing.

L36: You refer to FAO FRA 2015, while FRA 2020 already published and available.

The FRA 2020 figures have been updated on Line 36-37. The reference has also been updated on Line 389

L58: “an ideal and active carbon sink” What do you mean?

“Ideal and active carbon sink” means a cost-effective carbon sink. So, we changed the phrase as under.

Soils in juniper forests are also considered as cost-effective carbon sink and conserving this type of forest is imperative for carbon sequestration.

L83: please use “estimate” instead of “calculate”

Done

L111: Please clarify what does it mean: “The map data was extracted to excel to assess the study area in hectares”.

The sentence has been rephrased by the author as under:

The area for each class was calculated in ArcMap and the data was further analyzed by using MS Excel.

L114: “living biomass, shrubs” – shrubs are also “living biomass”, do you mean “above-ground live biomass of trees, shrubs”?

Yes, it means “above ground live biomass of the tree”. The change has been incorporated. 

L118: “non-availability of trees”, do you mean “absence”?

Yes, phrase “non-availability of the tree” has been replaced with “absence”.

L130: “which is why”

The sentence containing the above phrase has been rephrased as:

Shrubs and litter samples were collected as per their availability, and their location data points are displayed in Figure 3 and 4.

L143, 147: The equations do not consider that above-ground biomass includes branches, while stem volume not.

The equation covers the overall tree, including branches, and not just the stem volume.

L177: “Five samples were collected from different clusters”. It is not clear if five sample collected from every sample plot or five sample totally for the region.

The sentence and the numbers have been corrected. A total of six soil sample were collected, one from each cluster.

A total of Six soil samples were collected, one from each cluster. Bulk density of each soil sample was calculated in g/cm3.

L211: “training points” – did you use for the validation same points than were used for training the maximum likelihood classification algorithm?

No. The points used for the accuracy assessment were taken separately and are totally different from the training pixels used for the maximum likelihood classification algorithm.

L221: “plot no 14”

The change has been made to “plot no 14”

L224: “10.01 cm” One cannot measure tree diameter with such precision

Very true. The reviewer is right in this respect however, this value is not the diameter of one tree. This value is the average diameter of all the trees in that respective plot. Thus, the value in such manner is presented.

Table 4: “Sr. No; No of Trees” – do you mean “plot ID” and “number of trees”?

The change has been made in Table 4

Table 4: “Average Diameter (cm); Basal Area m2” Please put units (m2) uniformly in brackets

The change has been made in table 4

Figure 9-11: There is no sense for me to make tree separate predictions, one (mean) would be enough.

The authors also want to present the discrepancy or similarity that may arise in the predicted values using all the allometric equations. Although it is correct that one graph is enough for the presentation. For this purpose and as per the suggestion of other reviewers, we have combined Figure 9-11 in one Figure.

L312: “Felling trees for burning purposes” – Harvesting fuel wood?

Done

L328: “very less”, maybe “substantially less” or “much less”

Done

L350: “above ground, …, shrubs”. Do you mean “above-ground live tree biomass, …, shrubs”

Yes, the change has been incorporated.

Reviewer 3 Report

This paper evaluates the amount of carbon stock in an old juniper forest of Ziarat, Balochistan. The novelty of this paper consists in the forest type and location. Otherwise, the methodology was a standard method used generally for carbon stock evaluation. This study is a kind of necessary test of the generally used method applied for different ecosystems.

The English quality is good, and the paper reads well. However, the style is very didactic, reads like a scholar work. I think that there are many repetitions in the text that can be avoided. Also, there are some evident mentions, obvious in a scientific method, like in line 178: “Each soil sample was weighed, placed in bags, labelled and taken to the lab for analysis.”

I found the discussion interesting and presents well the difficulties of this study. But there are repetitions between discussion and conclusion. I found also a gap between discussion and the first part of the paper. The first part is very technical, containing a large amount of numbers. But the discussion does not interpret much the findings from the first part.

Caption of figures 2 to 4 is rather “location of tree plots” than “number of three plots”

Figure 5 : It is not the main subject of you paper, you can delete this figure. The text is enough.

Figures 9 to 11 : It is difficult to compare values. I suggest you to represent the values as three different lines in the same figure.

Line 165 : Figures 9 -11

Line 207: “shows”

Figure 8: What does “245” mean in the upper right corner?

Author Response

Response to reviewer’s comment on “Monitoring Carbon Stock and Land Use Change in 5,000-year-Old Juniper Forest Stand of Ziarat, Balochistan, through a Synergistic Approach”

The authors are grateful to the anonymous reviewers and editor for their positive assessment and the constructive comments. Efforts are made to address most of their comments and recommendations to further improve the quality of work presented in this manuscript.

Following the recommendations/comments of reviewers, authors made some major changes (in red) in the manuscript as given below:

Anonymous reviewer # 3

This paper evaluates the amount of carbon stock in an old juniper forest of Ziarat, Balochistan. The novelty of this paper consists in the forest type and location. Otherwise, the methodology was a standard method used generally for carbon stock evaluation. This study is a kind of necessary test of the generally used method applied for different ecosystems.

The authors are thankful to the reviewer for a nice feedback. It is true that the methodology used is a standard for the carbon stock evaluation and which is why the authors have used variegated equations to make the study more interesting.

The English quality is good, and the paper reads well. However, the style is very didactic, reads like a scholar work. I think that there are many repetitions in the text that can be avoided. Also, there are some evident mentions, obvious in a scientific method, like in line 178: “Each soil sample was weighed, placed in bags, labelled and taken to the lab for analysis.”

In response to the reviewer’s feedback, the text on Line 178 has been modified. The authors have tried to limit the repetitions of words and ideas to the minimum.

I found the discussion interesting and presents well the difficulties of this study. But there are repetitions between discussion and conclusion. I also found a gap between discussion and the first part of the paper. The first part is very technical, containing a large amount of numbers. But the discussion does not interpret much the findings from the first part.

The paper has been through extensive revision by addressing the comments from all three reviewers and we hope that significant improvement have been made in the manuscript.

Caption of figures 2 to 4 is rather “location of tree plots” than “number of three plots”

Done

Figure 5: It is not the main subject of your paper; you can delete this figure. The text is enough.

The Figure has been deleted and text has been added.

Figures 9 to 11: It is difficult to compare values. I suggest you represent the values as three different lines in the same figure.

The Figure 9 to 11 have been combined and the values have been represented in lines.

Line 165: Figures 9 -11

Done

Line 207: “shows”

Done

Figure 8: What does “245” mean in the upper right corner?

The author does not see any “245” in Figure 8. The point is not clear

Round 2

Reviewer 2 Report

All fine from my point of view besides of equations in lines 211-219, which need to be clarified.

217-219: Authors wrote “Total carbon in tree was assessed using the following equation. Total Carbon(kg) = Biomass (kg) x0.47”.

In contrast, in lines 201-213 authors wrote: “Stem Volume = B kg / WD / 100, where B is Biomass and WD is Wood Density”.

Please be careful with the terms used “total carbon in tree” should include carbon in stem (or trunk), branches, foliage and roots.

Stem volume and wood density are only related to the stem (trunk) and do not consider roots, foliage and branches.

These equations or terms use do not fit one another.

Author Response

Response to reviewer’s comment on “Monitoring Carbon Stock and Land Use Change in 5,000-year-Old Juniper Forest Stand of Ziarat, Balochistan, through a Synergistic Approach”

The authors are grateful to the anonymous reviewers and editor for their positive assessment and the constructive comments. Efforts are made to address most of their comments and recommendations to further improve the quality of work presented in this manuscript.

Following the recommendations/comments of reviewers, authors made some major changes (in red) in the manuscript as given below:

Anonymous Reviewer 2 (Round 2)

All fine from my point of view besides of equations in lines 211-219, which need to be clarified.

217-219: Authors wrote “Total carbon in tree was assessed using the following equation. Total Carbon(kg) = Biomass (kg) x0.47”.

In contrast, in lines 201-213 authors wrote: “Stem Volume = B kg / WD / 100, where B is Biomass and WD is Wood Density”.

Please be careful with the terms used “total carbon in tree” should include carbon in stem (or trunk), branches, foliage and roots.

Stem volume and wood density are only related to the stem (trunk) and do not consider roots, foliage and branches.

These equations or terms use do not fit one another.

The text on line 209 to 2019 are rephrased as under:

Fraction of carbon in the above ground living biomass can be assessed using the following equation adopted from FAO [31].

Total Carbon in above ground biomass (kg) = 0.475 x Above ground Biomass (kg)

Where the above ground biomass is assessed using the allometric equations listed in Table 2 The above ground living biomass volume formula was derived from the stem biomass formula of FAO [32] as mentioned below

Biomass (kg) = Vs * WD *1000

where Vs is stem volume and WD is wood density. Similarly, tree volume can be calculated by using following equation:

Biomass (kg) = Vt * WD *1000

Vt = Biomass/WD*1000

where Vt is tree volume and WD is wood density